# How Can We Do Citizen Science Better? A Case Study Evaluating Grizzly Bear Citizen Science Using *Principles of Good Practice* in Alberta, Canada

**DOI:** 10.3390/ani12091068

**Published:** 2022-04-20

**Authors:** Courtney Hughes, Krista Tremblett, Justine Kummer, Tracy S. Lee, Danah Duke

**Affiliations:** 1Alberta Environment and Parks, Government of Alberta, 9607 Shand Avenue, Box 239, Grande Cache, AB T0E 0Y0, Canada; krista.tremblett@gov.ab.ca (K.T.); justine.toppin@gov.ab.ca (J.K.); 2Miistakis Institute, Mount Royal University, Rm U271 Mount Royal University, 4825 Mount Royal Gate SW, Calgary, AB T3E 6K6, Canada; tracy@rockies.ca (T.S.L.); danah@rockies.ca (D.D.)

**Keywords:** citizen science, conservation, evaluation, grizzly bear, government, principles

## Abstract

**Simple Summary:**

Citizen science offers an excellent opportunity to engage the public in scientific data collection, educational opportunities, and applied management. However, the practicalities of developing a citizen science program, from generating ideas to developing tools, implementing programming, and evaluating outcomes, are complex and challenging. To address challenges and provide a foundation for practitioners, scientists, and the public, the Government of Alberta developed a set of citizen science principles. Here, we use these principles as an evaluative framework to assess the outcomes of the GrizzTracker program, which was developed to help inform provincial species-at-risk recovery efforts. While the program experienced some successes, we identified challenges, including skepticism from the scientific community about the utility of citizen science and a lack of program leadership, staff capacity, and funding needs for long-term implementation. Reflecting on the principles, we provide policy recommendations that future citizen science programs can consider.

**Abstract:**

Citizen science offers an excellent opportunity to engage the public in scientific data collection, educational opportunities, and applied management. However, the practicalities of developing and implementing citizen science programming are often more complex than considered. Some challenges to effective citizen science include scientists’ skepticism about the ability of public participants to rigorously collect quality data; a lack of clarity on or confidence in the utility of data; scientists’ hesitancy in engaging the public in projects; limited financial commitments; and challenges associated with the temporal and geographic scales of projects. To address these challenges, and provide a foundation upon which practitioners, scientists, and the public can credibly engage in citizen science, the Government of Alberta developed a set of citizen science principles. These principles offer a framework for planning, designing, implementing, and evaluating citizen science projects that extend beyond Alberta. Here, we present a case study using these principles to evaluate GrizzTracker, a citizen science program developed to help inform provincial species-at-risk recovery efforts. While we found that GrizzTracker applied each of the six principles in some way, including successful public engagement, strengthened relationships, and raising public awareness about northwest Alberta’s grizzly bears, we also identified a number of challenges. These included ongoing skepticism from the traditional scientific community about the utility of citizen science and governance challenges related to program leadership, staff capacity, and funding. By using the principles as a guideline, we provide policy recommendations for future citizen science efforts, including considerations for program design, implementation, and evaluation.

## 1. Introduction

Public participation in scientific research, also known as citizen science, has proliferated over the last two decades [1,2,3,4]. Increasingly, scholars and practitioners are recognizing the potential for citizen science to generate large-scale datasets at various spatiotemporal scales and support collaboration, relationships, and education among the public, scientists, policy-makers, and others [5,6].

Within conservation biology and environmental research, citizen science has contributed knowledge to a range of scientific, policy, and stewardship efforts. These include assessing vulnerable species [7], documenting migratory bird range shifts [8], recording urbanization effects on amphibians [9], monitoring lake water quality [10], reporting roadside animal carcasses to understand collision risks [11], integrating knowledge systems to understand freshwater mussel health [12], informing invasive marine species management [13], and reporting human–wildlife conflicts [14,15]. Through comprehensive design, including various social, political, and financial implementation supports, citizen science can both improve the public’s scientific literacy through co-learning and knowledge sharing [3,16,17] and enhance the relevancy and legitimacy of scientific research and applied management [3,18].

However, the scientific community remains reluctant to accept citizen science as a valid method of investigation, method of data generation, or approach to applied management [19,20,21,22,23]. There is a lack of understanding or appreciation by scientists of its value due, in part, to skepticism about the public’s ability to collect reliable data for applied decision-making [24,25,26]. Further, the amount of time and effort required for scientists to recruit, train, and retain volunteers [24,27,28] over the course of a research project is daunting, particularly if scientists are uncertain that they will achieve project outcomes [22,26,29]. Lastly, the rapid growth of citizen science presents a challenge, with diverse aims and applications contributing to a lack of cohesion in the field [30,31]. In this context, the lack of clear principles to guide the professional practice of citizen science adds to this hesitancy and skepticism. 

To address these challenges, government and non-government agencies are developing policies and principles to clarify and guide the development, implementation, and evaluation of citizen science programming. This includes identifying appropriate roles for researchers, practitioners, and participants and appropriate applications of citizen science in conservation contexts [31,32,33]. However, there is little documentation available on how guiding principles are practically applied to citizen science programming and what effect this has on achieving outcomes [18,31]. Here, we present a practitioner’s perspective on how, and to what extent, a set of guiding principles are applied to a citizen science program in a conservation context. In the following sections, we: (1) describe the development of guiding principles for citizen science in Alberta, Canada; (2) evaluate how the principles were applied to a citizen science program developed to help inform species-at-risk recovery efforts; and (3) identify the potential for principles to guide the development, implementation, evaluation, and sustainability of future citizen science programs.

## 2. Citizen Science Principles of Good Practice

Recognizing the potential for citizen science to contribute to our knowledge of environmental change, the Government of Alberta’s Department of Environment and Parks (AEP), together with the non-profit research organization the Miistakis Institute, developed the *Citizen Science Principles of Good Practice* (hereafter Principles) [1] to guide the design and application of citizen science (Table 1). The AEP’s mandate is to support the conservation and protection of the environment, including fish and wildlife species, among other areas. The Miistakis Institute was engaged to support the AEP in the development of these Principles. The Principles were developed collaboratively with the citizen science community in Alberta. In doing so, the Principles capture the knowledge and lessons learned from citizen science researchers and practitioners across the province [34]. The Principles serve as a foundation and catalyst to elevate the practice of citizen science as a legitimate means to inform policy decisions and the application of environmental conservation for the AEP. 

We use the Principles post-hoc as an evaluative framework to understand the successes and challenges of the GrizzTracker program [14,35] and provide recommendations for future policy direction and citizen science programming. 

## 3. Grizzly Bears and Citizen Science

Alberta grizzly bears (*Ursus arctos*) are a threatened species, with a provincial recovery plan implemented by the AEP identifying key strategies to address human-caused bear mortality and habitat loss [36]. In northwestern Alberta, an area designated Bear Management Area 1 (BMA 1; Figure 1) covers an area of approximately 40,000 km^2^ of the boreal forest. The area comprises extensive oil and gas developments, forestry operations, agricultural land (i.e., livestock and crops), recreational areas, and various smaller communities and rural farmsteads. A key component of recovery across this busy landscape is to engage the people living alongside bears to mitigate conflicts with them and secure habitat for future bear populations [37,38]. This includes supporting the scientific monitoring and reporting of bear observations and human interactions. 

Until recently, little was known about BMA 1’s population size and distribution [14]. However, people across BMA 1 would often report opportunistic grizzly bear sightings to government staff via phone calls, text messages, emails, or handwritten notes. These reports were often incomplete and lacked key details, including geo-referenced location, number of bears observed, bear activity, time of day, and observer effort. Recognizing the role that people can play in local bear management, AEP staff together with the Miistakis Institute worked with the Northwest Grizzly Bear Team to develop the citizen science program GrizzTracker (Figure 2) (see [14,35]).

In developing GrizzTracker, local AEP staff leading grizzly bear recovery efforts identified that not only would engaging citizens in grizzly bear science contribute much-needed data to support management, but also would also help raise awareness of the scientific methods used to study grizzly bears and show how data are used in policy decisions [40]. Further, AEP staff understood that engagement and collaboration with citizens would be an important opportunity to build open, trusting relationships and share information for the purpose of grizzly bear recovery [14,41].

Following a series of meetings amongst the Northwest Grizzly Bear Team, a unique smartphone application was developed that can record real-time grizzly bear observations on Android smartphones and iPhones using simple drop-down menus and photo uploads. The application also automatically collects anonymously geotagged locations (i.e., GPS-based points in 15-min intervals) of citizen scientists while the application is turned on and running in the background of the phone in order to measure observer effort in the field. The automatic collection of observer effort (i.e., the geographic and temporal distribution of observers across an area) is a novel feature of the smartphone application. These data are required for scientists to better understand where bears are being detected relative to where people are using the landscape, and in turn be useful in management decisions (i.e., not observing a bear in a certain area is just as important as observing a bear). Additionally, the program includes a website that provides educational materials on grizzly bear biology and ecology, scientific methods, land use management, and bear safety and conflict mitigation to help raise awareness and develop knowledge for citizen participants and broader public audiences.

## 4. Evaluative Framework: The Citizen Science Principles of Good Practice

To assess whether and how GrizzTracker followed the Principles, we reviewed Kelly et al.’s [18] citizen science evaluative framework and Kieslinger et al.’s [42] evaluation criteria to develop evaluative questions and indicators (Table 2) [14,43,44,45]. Then, using our expert judgment based on first-hand experiences and documentation during GrizzTracker’s development, we critically reviewed how the program performed using our criteria. Broadly, while we found that GrizzTracker applied each of the six principles in some way, including successful public engagement, strengthened relationships, and raising public awareness about northwest Alberta’s grizzly bears, we also identified a number of challenges. These included ongoing skepticism from the traditional scientific community about how useful citizen science data would be to grizzly bear recovery and persistent issues related to program leadership, staff capacity, and funding needs.

## 5. Discussion

Given the rapid growth in the popularity of citizen science, a standard set of principles can help guide the development of citizen science programming, implementation, and evaluation for researchers, decision-makers, and practitioners [31]. We used the *Citizen Science Principles of Good Practice* as an evaluative framework to assess how the GrizzTracker program supports grizzly bear conservation. While GrizzTracker was found to broadly and conceptually adopt most of the principles, the practical application of specific principles varied.

### 5.1. Successes

We clearly outlined the purpose of data collection (Principle #2), which was to complement our understanding of grizzly bear movements across the landscape specific to where people were using that same landscape. In turn, this would help mitigate and reduce conflict as well as support habitat management. GrizzTracker data were not used for regulatory purposes (enforcement actions or access restrictions), and quality control and assurance methods were designed in relation to the data’s later utility (Principle #2). The smartphone application was specifically designed to enable participants to document grizzly bear observations and contribute to data quality control by documenting observer effort (Principle #2 and #5) [35]. The GrizzTracker application was developed using open-source technology and was promoted through our Northwest Grizzly Bear Team, through the networks of current participants, and via the website and social media. This demonstrated openness and transparency and promoted learning opportunities (Principle #3 and #5). As a result, the smartphone application has been shared and used by other road ecology conservation projects in Canada, e.g., [11]. Additionally, GrizzTracker provided a broad range of information and educational opportunities for participants and the public to learn about bear ecology, scientific methods, bear safety, and public policy (Principle #4 and #5). For citizen science to be successful, engagement and collaboration with multiple audiences, including intended citizen scientists, are key [46]. We also developed trust amongst the Northwest Grizzly Bear Team and citizen science participants—basing relationships on open communication and respect and sharing knowledge and expertise, supported by the program design (Principle #3 and #4). While this required numerous meetings, we ultimately developed a common understanding of the problem, needs, and process to achieve success [14]. Working together helped strengthen relationships and deepen confidence in the science team, while the team itself learned the benefits of taking a non-traditional approach to collecting data, as well as address cynicism towards citizen science. We also found that active engagement in meaningful citizen science encourages participants to recruit other citizen scientists from their social networks, thereby expanding data collection and educational opportunities (Principle #4 and #5) [40,45].

### 5.2. Challenges

GrizzTracker was only able to partially fulfill the Principles, including inconsistent use of the application by participants when observing bears and, importantly, when not observing bears, which reduced the effectiveness of collecting observer effort (Principles #2 and #4). We partly attribute this to how difficult grizzly bears are to see, leading to relatively rare observations, coupled with participants forgetting to turn on the application or not truly understanding the need to collect observer effort data for use in recovery planning and management. In hindsight, it may have been more beneficial for participants to record all wildlife sightings rather than just grizzly bears (i.e., broaden GrizzTracker to include other species) to encourage consistent use. We also did not allocate systematic roadway surveys to participants, given that many BMA 1 roads are used for industrial and forest harvest activities. Thus, it was not convenient nor feasible (in terms of, e.g., employee work hours and safety precautions) for the employees to travel elsewhere, and so the road network was not evenly surveyed. However, we do note that other citizen science projects report opportunistic road network observations without accounting for sampling effort [16,47].

We also found that a lack of sustainable program coordination, including staff dedicated to managing check-ins and outreach with participants, hindered our ability to effectively communicate updates (Principle #4 and #5). Steenweg et al. [35] found that GrizzTracker participants wanted more frequent updates related to their data submissions and how data was being used. The lack of financial resources to hire staff dedicated to the program, coupled with poor support from senior management for citizen science programming, contributed to communication breakdowns.

There were also limitations in recruiting a broad diversity of participants, including those from Indigenous communities and the agriculture sector (Principle #5). We note that while petroleum industry and forestry employees were keen to participate, this is likely reflective of their own company mandates and requirements to follow legislation and regulations linked to grizzly bear recovery and staff safety. However, we are not suggesting that Indigenous communities or agricultural landowners were uninterested in grizzly bears. Rather, despite our best efforts to engage these audiences (i.e., in face-to-face meetings explaining the project), they chose not to participate and thus represent a data gap. We strongly suggest that future programming accounts for targeting different audiences, which means considering the most appropriate engagement approaches, considering the complexity of values and preferences for how people want to engage in wildlife conservation and management, as well as taking the time to build open communication lines and strong relationships. We suggest that this could be more effective at developing relevant and meaningful citizen science approaches and tools tailored to the needs of the public and scientists (see, e.g., [37,38]). 

That said, we do note that this limitation is particularly important to resolve amongst Indigenous communities. Indigenous peoples may be reluctant to engage in scientific activities given the history of colonialism and extractive research practices that have resulted in significant harms [48,49]. Certainly, citizen science initiatives must seek constructive ways to engage Indigenous communities “not just as actors carrying out information tasks or data collectors or as stakeholders defining research questions but, rather, as legitimate knowledge holders, respecting that their knowledge originates from different knowledge systems” [50] (p. 507). Further, we suggest that the lack of participation from certain groups in developing and using GrizzTracker reflects a hidden power dynamic. This suggests to us that reflection is required on engagement processes—who is inviting whom to the table, who shows up and why, and whose voices are heard [38]. Indeed, our evaluation suggests that it is naive to expect that ‘engagement’ in citizen science is as simple as an open invitation for people to participate. 

Lastly, we found that the persistent hesitancy for some participants in using the GrizzTracker application was based on concerns of ensuring confidentiality and anonymity in their data submissions (Principle #5). This was despite our attempts to alleviate such concerns by clarifying who has data access and securing their privacy. In part, this may be an artefact of broader trust issues the public has in government or technology and the purposes for which their data will be used.

### 5.3. Recommendations

Our evaluation revealed that GrizzTracker was founded on a collaborative engagement process that attempted to account for diversity, inclusivity, and trust-building across participants as well as develop technically and technologically appropriate tools while considering the constraints of program sustainability. The GrizzTracker program is still operating and accessible to the public; however, there is currently no dedicated staff or funding to manage the program.

We suggest that, in order to ensure the success of GrizzTracker, clear policy direction be given. Assurances on data collection and quality, data ownership and sharing, and access to results at the local level require a legal and organizational framework [51,52,53]. Several guiding principles currently exist on different aspects of citizen science that may be useful to consider [34], in addition to the Principles we offer here. 

There also needs to be investment in the coordination and leadership of citizen science programming at all levels, including addressing skepticism among scientists, developing meaningful engagement processes, and removing barriers to volunteer recruitment and engagement [54]. In particular, developing meaningful engagement processes that include open, two-way communication lines and feedback opportunities can help build more respectful and trusting relationships between citizen scientists, governments, and researchers [55]. Lastly, citizen science requires investment in program evaluation in order to ensure that challenges, lessons learned, and successes are recognized and recorded to improve future applications [42].

While we acknowledge that it is not easy to develop a citizen science program that balances the social, cultural, financial, and, in our case, conservation management needs, i.e., there is no ‘silver bullet’ to fix every problem [31], following clear principles like the *Citizen Science Principles of Good Practice* can help mitigate or avoid some challenges. Overall, we suggest that the value of citizen science must be more clearly articulated to scientists, organizations (whether government or otherwise), and public participants in order to help ensure that programs continue to be developed in ways that are sustainable, relevant, and supported. We hope that this will lead to the formal recognition of how beneficial community engagement in scientific data collection can be for conservation decisions and applied management.

## 6. Conclusions

Our paper described the development of the *Citizen Science Principles of Good Practice* for Alberta, Canada and our evaluation of a citizen science program called GrizzTracker. While GrizzTracker experienced much success relative to the program design and targeted engagement, our evaluation revealed some persistent challenges. Notably, we found a reluctance on the part of the scientific community to accept citizen science as well as skepticism from the public about participating in such a program. Both hindered the effectiveness of GrizzTracker. We also found that the amount of time and effort required to operate a program of this scale, including recruiting, training, and retaining volunteers, highlights the need for dedicated financial investment, staff, and decision-making leadership and targeted engagement of various audiences. Finally, we found that the lack of cohesion when designing, implementing, and evaluating the citizen science program reinforced the need for guidelines such as the Principles presented here. It is our hope that the Principles, evaluative framework, and lessons we learned will be useful in guiding future citizen science endeavors.

## Figures and Tables

**Figure 1 animals-12-01068-f001:**
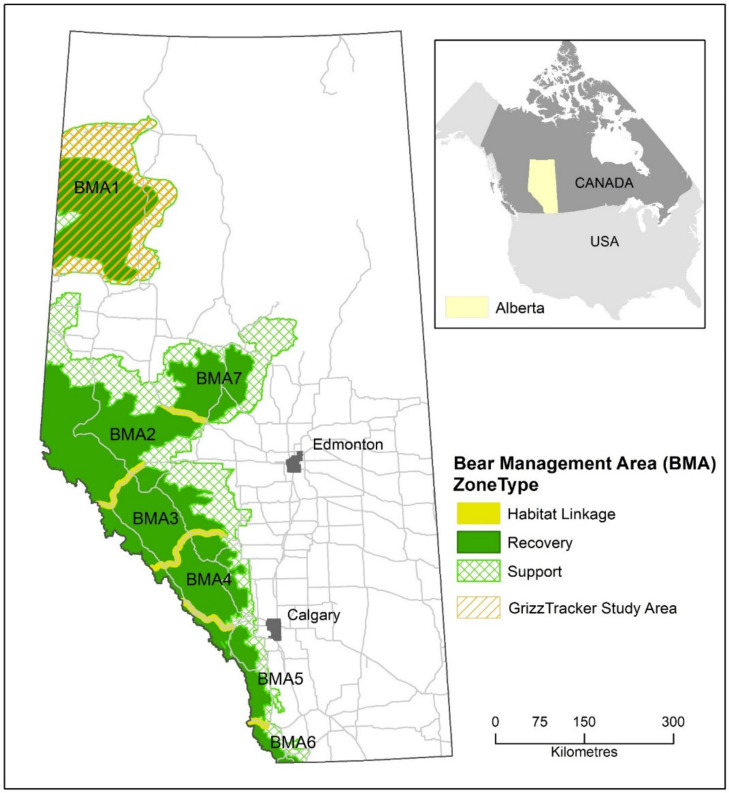
Bear Management Areas (BMAs) of Alberta, with identification of BMA 1 where the GrizzTracker program was pilot tested [39].

**Figure 2 animals-12-01068-f002:**
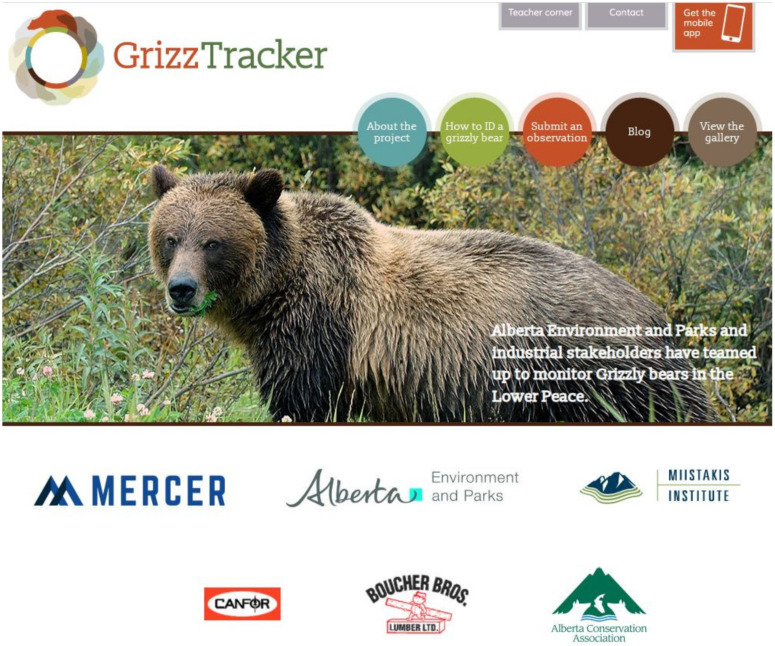
GrizzTracker website landing page (grizztracker.ca, accessed on 20 December 2021).

**Table 1 animals-12-01068-t001:** Six principles of good practice for citizen science [1].

Citizen Science Principles of Good Practice
1. Citizen science programs include a stated purpose and/or scientific outcome.
2. Citizen science data are fit to function and collected using standards and protocols appropriate to the intended purpose and/or scientific outcome.
3. Citizen science programs operate in an open and transparent manner.
4. Citizen science programs are inclusive and encourage active, meaningful, and productive citizen participation.
5. Citizen science programs are designed to provide benefits to all participants.
6. Citizen science programs take into consideration safety, legal, and ethical standards and guidelines.

**Table 2 animals-12-01068-t002:** Evaluation framework and results.

Principle	Evaluation Question(s)	Indicator(s)	GrizzTracker Application
1. Citizen science programs include a stated purpose and/or scientific outcome, such as generating new knowledge or informing conservation actions, environmental management decisions, or environmental policy.	What is the stated purpose and/or scientific outcome of the program? Are the scientific outcomes sufficiently clear?	Documentation of the program purpose, goals, and/or desired scientific outcomes (e.g., a program plan, a conceptual framework).	Scientific need: lack of a rigorous dataset on grizzly bears to use in recovery management planning. Program Goals: The Northwest Grizzly Bear Team identified the following goals: Improved grizzly bear population modeling; Improved human and wildlife safety; A platform for stakeholder learning; Deeper ecological literacy and a stronger sense of place.
2. Citizen science data are fit to function, collected using standards and protocols appropriate to the purpose and/or scientific outcome, and follow scientific practices in design, implementation, data quality assurance, data management, and evaluation.	How does the program design match the program purpose and/or scientific outcome? How does the program attend to quality assurance and quality control measures needed to produce rigorous, high-quality data? How do participant training and resources match the task (i.e., data collection)?	Documentation of the program design with specific outcome statements. Quality assurance and quality control measures for data (e.g., expert data verification). Participants provided with appropriate training and resources.	Intended purpose of the program: data would supplement grizzly bear monitoring, and public participation would increase scientific and bear awareness, knowledge, and skills. Data collection methods: a smartphone application was developed that supported standardized, automated, and rigorous collection of grizzly bear sightings, including a testing functionality to record observer effort. Quality control measures: Participants were asked to provide confidence in species identification. Unconfident records were removed from the analysis. Participants were provided with training sessions on the program and grizzly bear safety and conservation. A supporting website was developed that included a grizzly bear identification guide and quiz. Data were tested for bias and outliers during analysis. Similar attribute data were standardized between datasets. Post-program assessment: follow-up was limited due to capacity/resourcing limitations; however, an evaluation was conducted (see [35]). Additionally, there remains a lack of clarity on the utility of data for applied management.
3. Citizen science programs operate in an open and transparent manner and, where appropriate, project data, applications, and technologies are shared to encourage a culture of sharing and rapid innovation.	What data collection tools are being used and if new tools were designed could they be shared? Are data ownership and access rights clear and transparent? How is the project making data available? To whom? How is the project sharing results? With whom?	Data and results are shared with participants in suitable formats (e.g., data visualizations).	Open-source technology: a smartphone application was developed using open-source technology and shared/used in other citizen science programs. Data Sharing: Northwest Grizzly Bear Team members were provided with access to anonymized raw data via an administration portal. The project website enabled transparency and accessibility of results by displaying reported observations on a mapping tool after a two-day delay.
4. Citizen science programs are inclusive and encourage active, meaningful, and productive citizen participation.	What participant recruitment strategies were used to engage with a diversity of participants (e.g., gender, age, ethnicity)? To what degree are participants involved in project tasks (e.g., defining research questions, data collection, results interpretation, reporting)?	A diversity of participants are engaged throughout the project. Engagement in all aspects of the program, including defining the research questions and design, identifying objectives, data collection, analysis, and reporting/communicating outcomes.	Diversity and Inclusion: participation was initially limited. After a pilot phase, the program was opened up to broader public participation. Opportunities to engage: public participation focused on data collection, with program design, testing, and implementation performed by a representative stakeholder group.
5. Citizen science programs are designed to provide benefits to all participants, including citizens, practitioners, and researchers. Benefits include publishing research outputs, learning opportunities, personal enjoyment, social interaction, and contributing to scientific evidence. Whenever possible, with permission, participants should be acknowledged in project results and publications.	How did participant perspectives inform the program design? How are participants provided with ongoing opportunities for co-learning and sharing knowledge? How do both the researchers and participants benefit? How are participant contributions acknowledged?	Discussion of potential benefits to participants, including developing their skills or the creation of new knowledge, to help inform environmental decisions. Mechanisms to support co-learning and knowledge sharing exist. Participant satisfaction. Evidence of appropriate acknowledgement of participant effort (e.g., scientific publications, communications, products, public events).	Participants: a multi-stakeholder project team (the Northwest Grizzly Bear Team) with representatives from the Government of Alberta, energy and forestry resource sectors, environmental non-governmental organizations, and an academic research institute was established to identify program goals and benefits. Co-learning and sharing: The program provided a platform for shared learning between land managers and industry through collaborative program design, program implementation, and the sharing of findings. Acknowledgement: The participants were acknowledged during presentations and in published reports. Co-learning and sharing: The response rate to inquiries and the provision of feedback were delayed due to a lack of capacity/resourcing.
6. Citizen science programs take into consideration safety, legal, and ethical standards and guidelines surrounding copyright, intellectual property, confidentiality, data sharing agreements, and the environmental impact of any activities.	How does the project consider participant safety? How does the project consider existing policies and regulations that apply to the governance and management of data and information?	Protocols are established and participants trained on the protocols. Documentation of ethical research practices such as data sharing agreements.	Safety: Volunteer training provided a platform to provide educational information on safety, including human and bear conflict. Policy and Regulation: Data are owned by the Government of Alberta, although data and reports are shared with project partners and the broader public. Raw data for threatened species are not shared publicly. Observations are shared on a mapping tool but with a two-day delay to protect a species at risk.

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
