# Peer review of "How Can We Do Citizen Science Better? A Case Study Evaluating Grizzly Bear Citizen Science Using Principles of Good Practice in Alberta, Canada"

_animals, 2022, doi:10.3390/ani12091068_

Round 1

Reviewer 1 Report

The article critically and honestly evaluates a citizen science tool using existing guidelines developed by the Alberta provincial government. The information presented will be of value to the readership of the journal, as many researchers and managers are hesitantly considering citizen science solutions, as they do not know what this will actually entail both in terms of time involvement and the data generated. The value of the presented information is in a) presenting a citizen science case study, b) introducing the Alberta government’s 6 principle guidelines for good citizen science projects, and c) highlighting the importance of critically evaluating citizen science projects. The structure of paper is not typical of many manuscripts of the journal, but it is logically structured.  There are minor suggested edits, primarily regarding syntax and grammar, which I provide below.

General comments:

I would go through the manuscript carefully, trying to reduce the length of sentences where, possible (possibly rephrasing to remove words, or splitting sentences in half). The grammar is correct, but it is at times tiring to follow.

Abstract:

I believe that in the Abstract the reader should get a bit more information on what the GrizzTracker actually does, beyond what is mentioned in Line 37 (which is the same info as in the summary). Since the abstract is already quite long, I think effort should be made to tighten the preceding sentences a bit, so that the extra words explaining the app (a sentence – no more), should not add to the length.

Line 38: Each of the six principles… Strikes a bit odd the use of the definite article “the” here, as there has been no prior reference to them. May be say “each of the guidelines’ six principles…”

Introduction

Line 53: among instead of between

Line 57: A verb is used for all the examples of citizen science (e.g. documenting, reporting, monitoring), except for the first in the list – vulnerable species assessments. Add a verb here as well, as it strikes odd.

Line 73: with skepticism – replace with due or as well as (as appropriate)

Line 80: lack of cohesion in the field – unclear what this means. Rephrase for clarity

Line 105: among other areas – awkward phrasing/confusing – rephrase.

Line 107: unclear what the citizen science community means – the researchers/managers using it and/or the citizen scientists? Please clarify.

Table 1: Very important to present early on the six principles. I would prefer the text not to be center aligned.

Lines 113-114: Unclear here whether the project was indeed designed taking the 6 principles into account, and now self-evaluates whether these principles were met, or whether the GrizzTracker was designed before the 6 principles were proposed, and now post-hoc it examines how the project met them. Please calrify.

Line 121: avoid having the word area mentioned three times in one line if possible.

Line 128: delete “of”

Figure 1: Higher resolution is needed. Also, please add an inset of global map – highlighting maybe Alberta (or a map of Canada highlighting Alberta?)

Line 137: observer effort is a non-intuitive information that a citizen would mention in any report – strikes odd. Consider removing it.

Line 154: Unclear what the recording of observer effort entails.

Line 156: material instead of materials

Lines 161-163: As a reader, I would like at this point to read a bit more on what these two cited evaluation frameworks suggest. There are plenty of references, but since most of the readership will not be very familiar with social science qualitative evaluation protocols, it is worth expanding briefly here on the approaches used – beyond what is presented in Table 2.

Results:

While Table 2 (last column) presents the results of the GrizzTracker evaluation, I would like a brief summary of those findings to be in text format after the initial introductory paragraph of the section (Lines 160-164)

Table 2: I suggest it is presented in one page landscape format, with smaller font, and not center aligned text.

Discussion:

Line 170: Unclear what principles stand for

Line 187: How observer effort was estimated using the app is still unclear. This needs to be clarified earlier on in the manuscript.

Line 217: I understand just now that the App should have been switched off prior to a bear detection, and the route of the user anonymously tracked. But this should have been clarified earlier.

Line 278: Yes – but also important to propose ways that the end users (local people) actually trust these legal frameworks. As mentioned earlier, there is an inherent lack of trust towards governments. Maybe 1-2 sentences as to what could be done to change this?

Overall discussion comment – I would have liked to see here a step by step list of suggestions as to what you would have done different now (post-evaluation) were you to develop and implement GrizzTracker from scratch, having the knowledge you have. IS GrizzTracker still in use for instance? What measures have been adopted to improve it use? These would be practical suggestions that people considering citizen science projects would like to see.

Author Response

We would like to thank the Reviewer for their helpful advice in order to improve this manuscript. We have edited the Abstract to give more insight to the GrizzTracker program, but were mindful of word count as the Reviewer pointed out. We have made the suggested revisions to all line items identified.  We have clarified the post-hoc nature of the GrizzTracker evaluation. We have added an improved map with higher resolution oriented in Canada. We have clarified the section on observer effort. We are unsure what the Reviewer is requesting with regards to the suggestion for Lines 161-163. We think that the references and our explanation of our evaluative framework is clear, though if further revision is necessary request that the Reviewer please clarify. We have formatted the Table to landscape orientation. We have clarified the identified discussion section revisions, according to the Reviewer request. We have added clarification to the aspect of building trust and its relationship to citizen science. We reviewed the manuscript and the other Review comments, and in light of this has decided not to rewrite the recommendations as a listing. We felt our format and organization nicely reflects what we offer as recommendations, though we do appreciate the suggestion from the Reviewer.  

Reviewer 2 Report

Good report that will hopefully be useful to others. I was confused about line 93 where you "describe the development of guiding principles for citizen science in Alberta, Canada"

Surely your paper is not aimed only at citizen scientists in Alberta, as the rest of your paper demonstrates how your Principles can be used elsewhere, so I was confused about why that sentence was specific to Alberta. 

Perhaps you meant "describe the development of guiding principles for citizen science, using Alberta’s grizzly bears as a test case", or something similar?

The only other comment I have is that Figure 1 is very low-res, so hopefully you can use and better resolution image for the final version.

Author Response

We would like the thank the Reviewer for the advice given to improve our manuscript. We have addressed the revisions suggested and have edited and revised throughout for grammar, syntax and clarity, as well as an improved Figure 1. Thank you.